# Comparative Study on Microstructure of Mo/Si Multilayers Deposited on Large Curved Mirror with and without the Shadow Mask

**DOI:** 10.3390/mi14030526

**Published:** 2023-02-24

**Authors:** Xiangyue Liu, Zhe Zhang, Hongxuan Song, Qiushi Huang, Tonglin Huo, Hongjun Zhou, Runze Qi, Zhong Zhang, Zhanshan Wang

**Affiliations:** 1Institute of Precision Optical Engineering, School of Physics Science and Engineering, Tongji University, Shanghai 200092, China; 2MOE Key Laboratory of Advanced Micro-Structured Materials, No. 1239 Siping Road, Shanghai 200092, China; 3Department of Physics, Shanghai University, Shanghai 200444, China; 4National Synchrotron Radiation Laboratory, University of Science and Technology of China, Hefei 230029, China

**Keywords:** multilayer, shadow mask, curved substrate, magnetron sputtering

## Abstract

The Mo/Si multilayer mirror has been widely used in EUV astronomy, lithography, microscopy and other fields because of its high reflectivity at the wavelength around 13.5 nm. During the fabrication of Mo/Si multilayers on large, curved mirrors, shadow mask was a common method to precisely control the period thickness distribution. To investigate the effect of shadow mask on the microstructure of Mo/Si multilayers, we deposited a set of Mo/Si multilayers with and without the shadow mask on a curved substrate with aperture of 200 mm by direct current (DC) magnetron sputtering in this work. Grazing incidence X-ray reflectivity (GIXR), diffuse scattering, atomic force microscope (AFM) and X-ray diffraction (XRD) were used to characterize the multilayer structure and the EUV reflectivity were measured at the National Synchrotron Radiation Laboratory (NSRL) in China. By comparing the results, we found that the layer microstructure including interface width, surface roughness, layer crystallization and the reflectivity were barely affected by the mask and a high accuracy of the layer thickness gradient can be achieved.

## 1. Introduction

As the refractive indexes of all materials are close to one in the extreme ultraviolet (EUV) region, the single reflective layer coating can only be used in grazing incidence geometry. Multilayer coating can obtain high reflectivity at near-normal incidence in the EUV region [1]. With the development of EUV multilayers, the normal incident optical systems have been realized in many fields. Mo/Si multilayer is one of most promising combinations of EUV multilayers because of its excellent optical properties; it has been widely used in lithography, astronomical observations, plasma diagnostics, synchrotron radiation light sources and other fields [2]. At the wavelength of 13.5 nm, many research groups have been able to fabricate Mo/Si multilayers with the reflectivity close to 70% at near normal incidence [3,4].

With the development of EUV optical systems, the need for large numerical aperture (NA) optical systems has become more common [5]. Large NA optical systems not only require larger optical components, but also need optical components with complex and curved surface profiles. Compared to flat substrates, it is difficult to deposit a film on a curved substrate with precise control of period thickness. For large, curved mirrors with multilayer coating in actual applications, the incident angles at different locations on the mirror’s surface are different. In order to achieve good optical performance, the curved mirror needs a laterally graded multilayer [6]. The laterally graded multilayer needs precise control of multilayers thickness, and there are two main methods during the deposition. One is changing the speed profile of substrate motion [7,8,9,10,11,12] and the other is using the shadow mask to correct the period thickness of multilayers [13,14,15,16,17,18,19,20,21,22,23,24]. To reduce the thickness deviation on the whole mirror, they changed the speed profile of the substrate motion. Yu et al. reduce the thickness deviation of the Mo/Si multilayer onto a curved mirror with a clear aperture (CA) of 184 mm and radius of curvature (RoC) of 338.66 mm by DC magnetron sputtering with planetary rotation stages (PRS) below 0.09% [25]. Morawe et al. prepared gradient multilayers with a period thickness from 3 to 6 nm on a 240 mm long substrate. The periodic thickness error was less than 1% [26]. This technique requires a high degree of mechanical accuracy in the deposition system, which is hard to achieve. Other researchers choose to use the shadow mask to correct the thickness of the multilayer. Yulin et al. have optimized the layer thickness distribution by placing a specially-formed shadow mask near the cathode. Then, they achieved a homogeneity of ±0.1% on a 150 mm area and ±0.2% on a 300 mm area [27]. In 2009, Sassolas et al. used a curved mask to control the coating thickness deviation. They obtained a deviation which was lower than 0.7% on a 500 mm substrate [20]. Film uniformity is also sensitive to the distance between the substrate and target. In general, the further the substrate is placed, the better the uniformity of the film [28].

However, these studies mainly focused on the optical properties on large, curved substrates, but the effect of the shadow mask on the microstructure of multilayers has not been mentioned. In this paper, a set of Mo/Si multilayers have been deposited with and without the shadow mask on a large, curved substrate with an aperture of 200 mm. Then, a systematic comparison and analysis of layer structure, morphology, internal microstructure and optical performance of the multilayers have been presented. It was found that the use of shadow mask does not affect the properties of the multilayers mentioned above.

## 2. Experimental Details

Figure 1a shows a schematic diagram of the direct current (DC) magnetron sputtering system used in this study, based on a planetary rotation mode. During the deposition, the substrate is self-rotated with high speed to improve the uniformity of radial direction of multilayers, while the substrate passes sequentially through the Mo and Si targets to obtain the multilayer. The Mo and Si targets are square and placed symmetrically in the cylindrical vacuum chamber. The large, curved substrate with a diameter of 200 mm is mounted face down to the targets. 

Since the multilayer’s thickness on the large, curved substrate could not be measured directly, a substituted substrate was prepared to estimate the multilayer’s thickness at selected points on the surface. To reproduce the real curvature of the substrate as accurately as possible, eight points between the center of the mirror and the boundary were selected, and their tangent line was used to create a cross-sectional line as shown in Figure 1b. Here, the upper image shows the cross-section of the large, curved substrate, and the blue spots represent the selected points (the deposition area was beginning at point 1 and ended at the boundary). The eight selected points could be in one direction along the radial axis. To provide enough space for mounting silicon wafers on the mirror substitute, four points are put on each side as shown in Figure 1b. The lower image depicts the cross-section of the substituted substrate, and the red line denotes the silicon wafer. The angle between each silicon wafer and the horizontal line was corresponding to the angle between the tangent line of each selected point and the horizontal line. Two samples at different locations were chosen for comparison, one closer to the center of the substituted substrate at X = 47.5 mm and the other closer to the edge of the substituted substrate at X = 92.5 mm (X indicates the distance between the sample and the center of the substituted substrate). This was done to compare the microstructure of multilayers deposited with and without the shadow mask and exclude the effect from different locations on the substituted substrate.

In this paper, for obtaining Mo/Si multilayer with a transverse gradient distribution of periodic thickness, a shadow mask is used to control the thickness of the multilayers at different locations on the substrate. In the experiment, we fixed the shadow mask on the outer ring of the sample holder using screws. The shadow mask was fixed during the deposition while the substrate kept self-rotation with high speed as shown in Figure 2. The size of the substituted substrate in this work is the same as the large, curved substrate; both of them have a diameter of 200 mm. The masking area of the shadow mask we used is also about 200 mm in diameter. This technique could control the amounts of deposited particles by adjusting the mask ratio at different locations, and then obtaining different multilayer thicknesses at different locations on the large, curved substrate. 

In this work, multilayers consisting of 50 bilayers of Mo and Si were deposited by the DC magnetron sputtering onto 20 mm × 14 mm super polished silicon wafers, and the period range is from 6.97 nm to 7.07 nm. In all experiments, high-purity argon (99.999%) was used as the working gas, the background pressure was 6.4 × 10^−5^ Pa and the argon gas pressure during deposition was 0.133 Pa. The deposition parameters in the experiment are shown in Table 1.

## 3. Results

### 3.1. Multilayers Thickness Control on Large Curved Substrate

As mentioned above, the periodic thickness at different locations on the substrate needs to be regulated by the shadow mask. The expected normalized thickness profile is shown in Figure 3 marked with black squares. In this work, the expected thickness of the multilayers is calculated based on the principle of achieving a EUV reflectivity as high as possible. Due to the large variation of incident angle on the large, curved substrate, the period thickness needs to be changed according to the working angle at different locations on the whole large, curved substrate. In Figure 3, the thicknesses were all normalized by the multilayer period thickness at the outermost area. When the substrate is deposited without a shadow mask, the obtained multilayer periodic thickness distribution across the whole area is 7.09 to 7.00 nm from the center to the edge. With the correction of the shadow mask, the periodic thickness distribution across the substituted substrate changes from 6.97 to 7.08 nm from the center to the edge. The periodic thickness at each location on the entire substrate can be achieved within ±0.02 nm of the expected thickness, which corresponds to ±0.3% of the expected thickness at each location. 

### 3.2. Grazing Incidence X-ray Reflectivity (GIXR)

In order to know the multilayer structure of the samples, GIXR tests were performed on the samples using a Bede D1 X-ray diffractometer made in Germany in θ-2θ mode. Figure 4 shows the GIXR curves of the prepared multilayer samples at different locations. Based on the GIXR test results, the Bragg peaks in the test curves of the samples with and without the shadow mask at different locations of the substituted substrate are extremely sharp and the heights are basically the same, which means all of these samples have similar layer structure. 

To further understand the structural parameters of the multilayers at different locations with and without the shadow mask, the test curves were fitted by IMD software [29], and the fitted structural parameters are shown in Table 2 and Table 3. From Figure 4, the fitted curves agree well with the test curves, indicating that the fitted results are consistent with the actual layer model. From the fitting results, the width of the interface at each layer of the sample is similar at different locations on the substituted substrate with and without the shadow mask. The fitted results agreed with previously reported results [30].

### 3.3. Atomic Force Microscopy (AFM)

The roughness of the multilayer was measured by atomic force microscopy (AFM) using Dimension^®^ Icon™ made in Germany with a scanning area of 1µm × 1µm. In order to investigate the effect of the shadow mask on the surface roughness and to exclude other factors that may have an effect on the surface roughness, the samples on the substituted substrate with and without the shadow mask close to the center and the edges were tested separately, and the results are shown in Figure 5a–d. The surface roughness of all samples is lower than 0.2 nm (root mean square, RMS), which is similar to the results of others’ studies [31,32]. The surface roughness of the samples located at X = 47.5 mm and X = 92.5 mm deposited with shadow mask is 0.174 nm and 0.179 nm, respectively. The surface roughness of the samples located at X = 47.5 mm and X = 92.5 mm deposited without shadow mask is 0.181 nm and 0.179 nm, respectively. This indicates that the surface roughness of the samples at different locations is the same on the whole substituted substrate. The surface roughness of the sample is not influenced when the multilayers are deposited with the shadow mask. Meanwhile, the power spectral density (PSD) functions of these samples are compared, and the results are shown in Figure 5e. The PSD function can quantitatively express the roughness intensity distribution of the multilayers surface in different spatial frequency ranges. The PSD functions of the samples deposited with and without the shadow mask at different locations on the substituted substrate basically overlap. It is further illustrated that the surface roughness of the multilayer samples is not affected by the shadow mask or the location of the sample.

### 3.4. Diffuse X-ray Scattering

In order to further understand the effect of the shadow mask on the structure at the multilayers interface, Mo/Si multilayer samples were tested by diffuse X-ray scattering, mainly to characterize the rocking curves near the first Bragg peak. According to GIXR and AFM test results, it can be determined that the distance between the sample and the center of the substrate has no effect on the layer structure and morphology of the multilayers. Therefore, two samples at X = 92.5 mm deposited with and without the shadow mask were selected to investigate the influence of the shadow mask on the interfacial structure. Figure 6 shows the rocking curves near the first Bragg peak for the samples. From the test curves, the scattering curves of the two samples are similar in shape. However, the wings on both sides of the scattering curves of the sample deposited without the shadow mask are slightly lower in intensity. The scattering test curve only receives the effect of interfacial roughness. It is thought that the interfacial roughness of the sample deposited without the shadow mask is slightly lower. But this effect is not reflected in the fitting results of the GIXR test curve. This may be because the interface width in the GIXR fitting is the result of the combined effect of interfacial roughness and diffusion at the interface and the interdiffusion is dominating in this case. 

### 3.5. X-ray Diffraction (XRD)

In this section, the formation of crystallization in Mo/Si multilayers is discussed. Since the crystallization of the Mo layer is mainly influenced by the thickness of the Mo layer, and the fitting results of the GIXR test curve in the first section show that the thickness of the Mo layer is basically the same throughout the substituted substrate, only the samples at X = 92.5 mm deposited with and without the shadow mask were selected for comparison. The θ-2θ mode scan and grazing incidence X-ray diffraction (GIXRD) mode scan (incidence angle θ = 1°) were performed on these two samples using Bruker D8 X-ray diffractometer made in Germany, and the test results are shown in Figure 7. From the test results of the two mode scans, the crystalline intensity of the samples deposited with and without the shadow mask is basically the same. The test curves of the θ-2θ mode scan were fitted by Jade software (PDF#42-1120), and it was found that the peak mainly correspond to Mo (110) crystallites in both samples, which was also found in other studies [33,34]. Based on the width of the Mo (110) diffraction peak, the crystallite size of Mo was calculated using Scherrer’s formula [35],
(1)$$H=\frac{0.94\lambda }{Lcos\left(\theta \right)},$$
where *λ* = 0.1542 nm, *L* is the full width at half maximum determined for every peak in the spectrum and *θ* is the Bragg diffraction angle. The crystallite size in the film growth direction can be easily calculated using this equation, and the results of the crystallite size calculations for these two samples are shown in Table 4. The crystallite size of the two samples is basically the same. From a previous study [36], we also know that the size of the Mo crystallites in the growth direction is equal to the thickness of the Mo layer, which was also found in our study. Thus, it can be concluded that the shadow mask does not affect the crystallization of the samples.

### 3.6. EUV Reflectivity

The EUV reflectivity tests of Mo/Si multilayers were performed on a reflectometer at the National Synchrotron Radiation Laboratory (NSRL) in China. The reflectivity of the prepared multilayers was measured at 12–15 nm with a step size of 0.015 nm. To improve the spectral purity, 600 lines/mm grating was used as a monochromator in the test. From the above subsections, it is known that the layer structure, morphology and internal microstructure of the Mo/Si multilayers at different locations deposited with and without the shadow mask is basically the same. Therefore, only one sample was selected in each experiment using shadow mask or not and tested at the near incidence angle θ = 10.7°; the test results are shown in Figure 8. The test results show that both of the samples have a high reflectivity at 12–15 nm around 65%. This indicates that the shadow mask does not affect the EUV reflectivity. Meanwhile, the EUV reflectivity of the samples was calculated by the fitting results of the GIXR test curves in the 3.1 subsection. The calculated EUV reflectivity curve basically overlapped with the EUV reflectivity test curve, which verified the correctness of the GIXR fitting results.

## 4. Conclusions

The interfacial structure and the surface roughness of the Mo/Si multilayers deposited with and without a shadow mask were characterized by different test methods. With the correction of the shadow mask, the periodic thickness at each location on the whole substrate can be achieved within ±0.02 nm of the expected thickness, corresponding to ±0.3% of the expected thickness at each location. The surface roughness of the samples deposited with and without the shadow mask is the same on the whole substituted substrate, all lower than 0.2 nm. The crystalline intensity of the samples deposited with and without the shadow mask is basically the same, and the shadow mask does not affect the crystallization of the multilayers. Additionally, both samples deposited with and without the shadow mask have high reflectivity at 12–15 nm around 65%. It was found that the layer microstructure including interface width, surface roughness, layer crystallization and the reflectivity were barely affected by the mask and a high accuracy of the layer thickness gradient can be achieved. This work can provide useful guidance for the deposition of a large, curved multilayer mirror for EUV optical systems.

## Figures and Tables

**Figure 1 micromachines-14-00526-f001:**
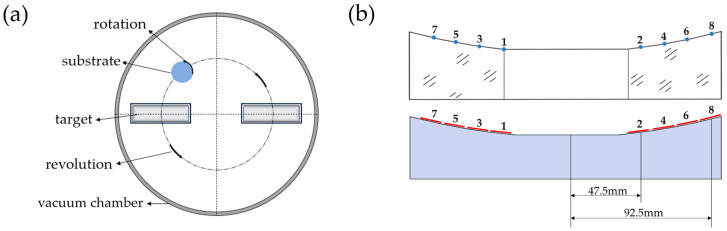
(**a**) Schematic of the direct current (DC) magnetron sputtering system used in this study; and (**b**) the substituted substrate (Numbers 1 to 8 refer to serial numbers of the samples).

**Figure 2 micromachines-14-00526-f002:**
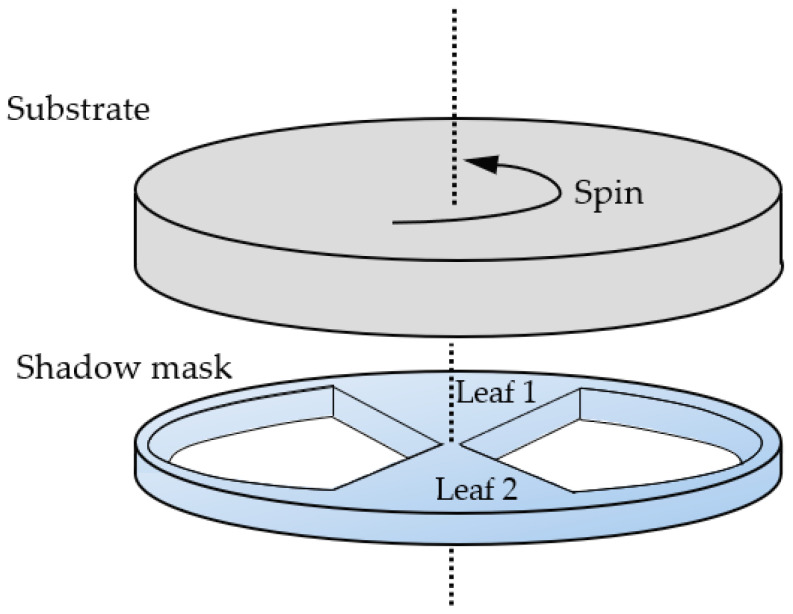
Schematic diagram of the geometric relationship between the substrate and the shadow mask.

**Figure 3 micromachines-14-00526-f003:**
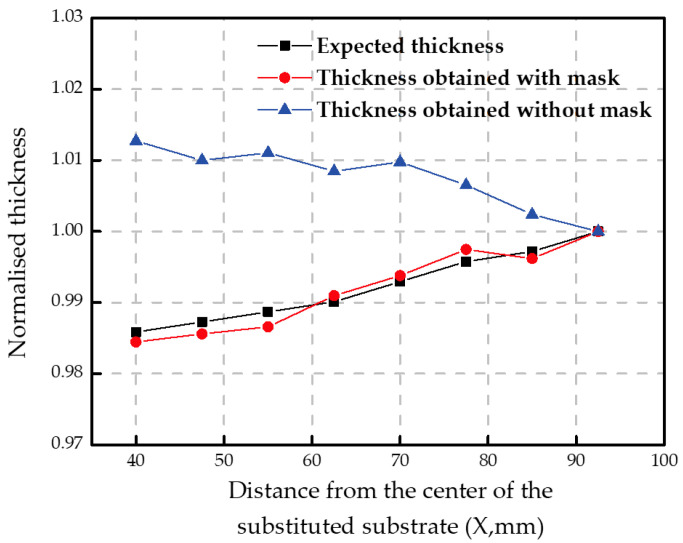
Normalized results for periodic thickness: expected thickness (black line), thickness obtained with the shadow mask (red line) and thickness obtained without the shadow mask (blue line).

**Figure 4 micromachines-14-00526-f004:**
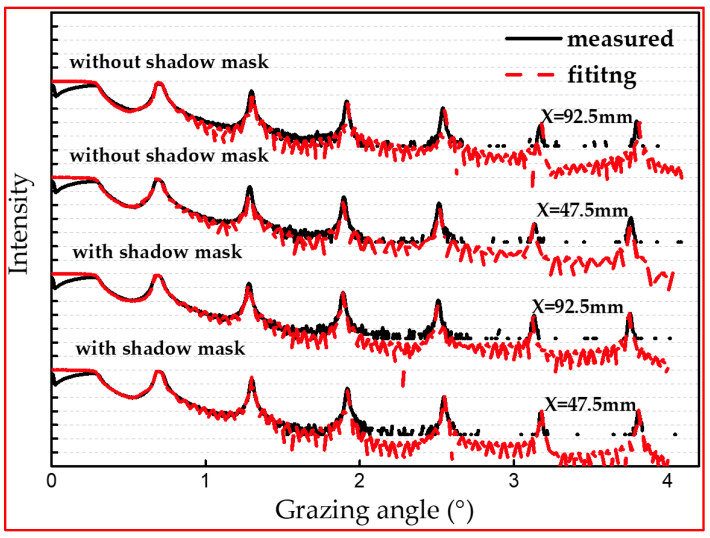
GIXR measurement results (black line) and fitted (red dots) GIXR curves of multilayers deposited with and without the shadow mask on different silicon wafers at different locations (X indicates their location coordinates).

**Figure 5 micromachines-14-00526-f005:**
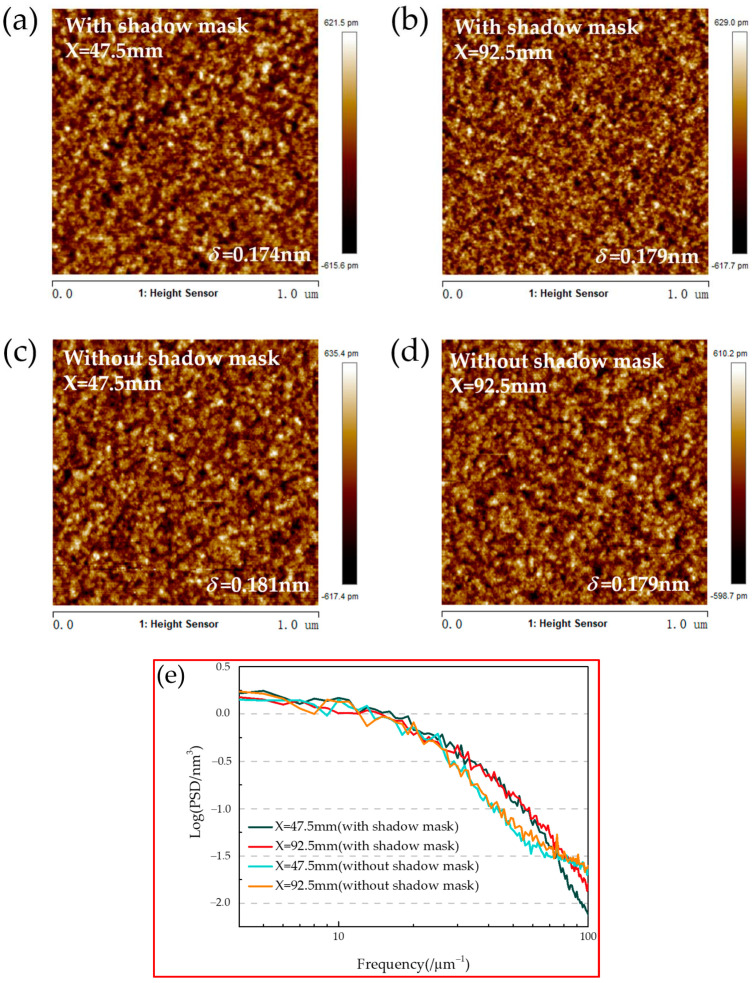
AFM picture with a scan width of 1 um of sample deposited with and without the shadow mask. (**a**) Sample at X = 47.5 mm deposited with shadow mask; (**b**) sample at X = 92.5 mm deposited with shadow mask; (**c**) sample at X = 47.5 mm deposited without shadow mask; (**d**) sample at X = 92.5 mm deposited without shadow mask; and (**e**) The PSD function obtained based on the AFM test plots function.

**Figure 6 micromachines-14-00526-f006:**
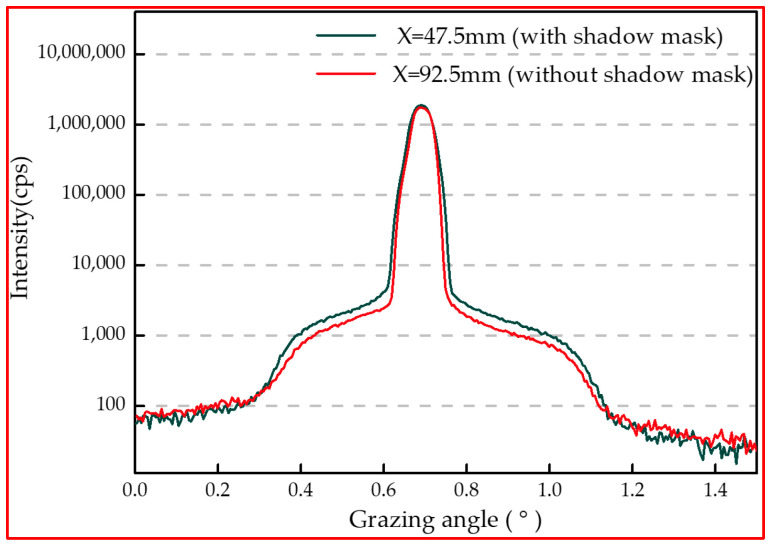
Rocking curves near the first Bragg peak at the same location with and without the shadow mask.

**Figure 7 micromachines-14-00526-f007:**
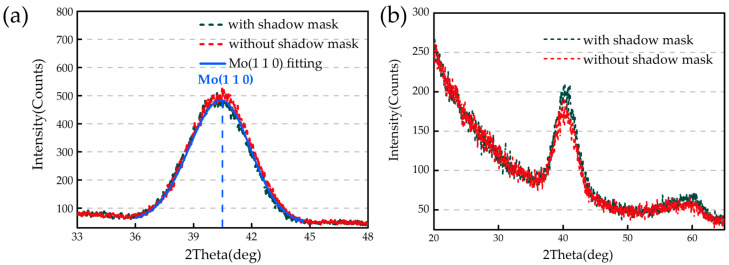
Crystallization test results. (**a**) θ-2θ mode scan; and (**b**) GIXRD.

**Figure 8 micromachines-14-00526-f008:**
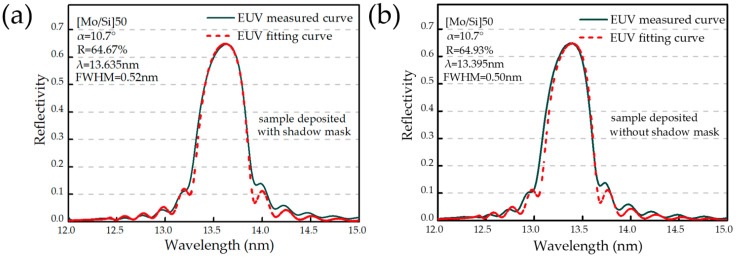
EUV reflectivity test results. (**a**) Sample deposited with shadow mask; and (**b**) sample deposited without shadow mask.

**Table 1 micromachines-14-00526-t001:** Deposition parameters in the experiment.

Deposition Parameters	
Number of period multilayers	N = 50
Period thickness	d = 6.97–7.07 nm
Working gas	99.999% Ar
Working gas pressure	0.133 Pa
Background pressure	6.4 × 10^−5^ Pa

**Table 2 micromachines-14-00526-t002:** Fitted results of Mo/Si multilayers deposited with shadow mask.

	X = 47.5 mm		X = 92.5 mm	
Layer	Thickness (nm)	Roughness (nm)	Thickness (nm)	Roughness (nm)
Mo	1.950	0.245	2.130	0.237
Mo-on-Si	0.806	0.316	0.803	0.320
Si	3.843	0.229	3.770	0.212
Si-on-Mo	0.386	0.428	0.400	0.425

**Table 3 micromachines-14-00526-t003:** Fitted results of Mo/Si multilayers deposited without shadow mask.

	X = 47.5 mm		X = 92.5 mm	
Layer	Thickness (nm)	Roughness (nm)	Thickness (nm)	Roughness (nm)
Mo	2.140	0.238	2.095	0.227
Mo-on-Si	0.802	0.320	0.788	0.318
Si	3.720	0.236	3.703	0.222
Si-on-Mo	0.409	0.442	0.388	0.436

**Table 4 micromachines-14-00526-t004:** Size of the Mo crystallite of the samples deposited with and without the shadow mask.

Sample	FWHM (deg)	2theta (deg)	Crystal Size (nm)
Deposited with shadow mask	4.12	40.515	2.15
Deposited without shadow mask	4.08	40.515	2.17

## Data Availability

Not applicable.

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
