# Peer review of "Comparative Study on Microstructure of Mo/Si Multilayers Deposited on Large Curved Mirror with and without the Shadow Mask"

_micromachines, 2023, doi:10.3390/mi14030526_

Round 1

Reviewer 1 Report

1. Many previous works have shown that different distance between substrate and target is easy to affect film uniformity. Please investigate the literature and analyze the differences between their works and this paper.

2. Please give details of the size of the substituted substrate and the shadow mask. How does the mask attach to the substrate?

3. Figure 3 lacks the unit for normalized thickness.

4. What is the basis for the expected thickness?

5. What kind of equipment is utilized for measuring the thickness of the layers on the substituted substrate?

6. The peaks analyzed in Figure 4 should be clearly marked.

7. ‘When the substrate is deposited without a shadow mask, the obtained multilayer periodic thickness distribution across the whole area is 7.09 to 7.00 nm from the center to the edge.’, How is the thickness change obtained?

8. How many samples do you fabricate for calibration of thickness? Please give the average thickness under different samples to provide better support for subsequent research.

9. Much literature is needed to prove the correctness of the analysis results such as GIXR, Diffuse X-ray scattering, XRD, and so on.

10. There are many grammatical mistakes in the article. For example, ‘the’ or ‘a’ is missing before many nouns such as ‘with and without shadow mask’, ‘ other field ’ should be replaced by ‘ other fields ’, ‘ two sets of Mo/Si multilayers does not match what was described earlier, etc.

Reviewer 2 Report

The paper presents a study on the influence of using a shadow mask to  control the thickness of a Mo/Si multilayer deposited on a curved substrate.

The researches carried out are thorough and led to interesting results.

However, the presentation of elements in the conclusion could be improved. Most elements in the conclusions suggest that the presence or absence of the shadow mask has the same effect on the structure of the multilayer, only one sentence in the middle shows a difference and then the actual conclusion that the shadow mask controls effectively the periodic thickness of multilayers and ensures the quality of multilayers does not immediately look justified.

The English language in the paper itself is good, but the same cannot be said about the abstract, where also the sentence structure should be improved.

Reviewer 3 Report

INTRODUCTION

The description, state of the art, and in general the introduction content are clear. The reading of the section is simple; however, there is no approach to impact real sectors. What kind of companies will be impacted? 

MATERIALS AND METHODS

Please increase quality of figure 1.

In general, the experimental procedure is well described with a high detail level. I recommend to include a table with deposition parameters.

RESULTS

At the technical and research level, the article is interesting. This involves several microstructural techniques of characterization; however I think that a HRTEM images and analysis to contrast against GIXRD could be very useful. Please consider to highly improved the quality of the manuscript.

Despite the several amount of characterization techniques used in the work, the lack of analysis in some sections is relevant. For instance, there are no investigation or calculations of microdeformations that can be done with XRD. This is useful to looking for a relationship with microstructure, crystallite size, roughness. 

It is very outstanding not to find nanoindentation tests!!! Why not?

It would be interesting to analyze the results with similar previous ones.

GENERAL COMMENT

Please increase the quality of all images.

Round 2

Reviewer 1 Report

Accept as it is